# Efficacious Utilization of Food Waste for Bioenergy Generation through the Anaerobic Digestion Method

Preethi Muthu [1], Gunasekaran Muniappan [1] and Rajesh Banu Jeyakumar [2],*

[1] Department of Physics, Anna University Regional Campus, Tirunelveli 627007, Tamil Nadu, India
[2] Department of Biotechnology, Central University of Tamil Nadu, Neelakudi,
Thiruvarur 610005, Tamil Nadu, India
* Correspondence: rajeshces@gmail.com; Tel.: +91-9444215544

**Abstract:** Energy fuels retrieved from biomass utilization are considered to be an economically and environmentally friendly source. In this day and age, bioenergy provides an alternative option to replace traditional fossil-based energy to accomplish energy demand with fewer greenhouse gas emissions into the environment. A huge amount of food waste is produced every year due to mass ethnographic activities. Their potential has been underused and this has led to waste ending up in the garbage. Bioenergy production by anaerobic digestion of cheap substrate provides an effectual approach to cope with this issue. The hydrolysis stage during anaerobic digestion is enhanced by various pretreatment methods, where the disintegration of the waste substrate leads to the enhancement of soluble organics and eases the production of bioenergy. The present review focuses on state-of-the-art knowledge about food waste, its utilization, and its valorization by the action of pretreatment, thereby enhancing anaerobic digestion. Additionally, this review further focuses on the major challenges during the pretreatment method and future recommendations.

**Keywords:** hydrolysis; valorization; substrate; exploitation; biofuel





## 1. Introduction

Globally, a major apprehension is the energy requirement and the environmental effluence. Thus, there is a need to develop eco-friendly and potential technologies that fulfill the energy demands and maintain the ecological balance caused due to the increase in urbanization and the population [1]. The major hitch that is associated with the utilization of fossil fuel is its negative impact on the environment which damages the health of an ecosystem. Moreover, the increase in industrialization and population has led to the increased usage of natural resources and thus an abundant generation of waste occurs that troubles the natural environment. Thus, a technology called waste to energy has been adopted to conquer the troubles in maintaining sustainability in the environment. This technology produces an abundant amount of energy and heat out of waste and reduces the usage of non-renewable fuels and thus mitigates greenhouse gas emissions [2]. Municipal and industrial waste is considered to be an advantageous source of energy since the disposal of waste occurs in a safe manner [3]. Out of different types of waste, food waste (FW) is considered to be efficient due to its abundant generation and higher organic content that eventually helps in energy production.

The present review encompasses the characterization of FW and its management strategy with a major emphasis on the technical features of disposal and energy retrieval options. The review begins with the organic matters present in FW and the major impeding components present in FW. Further, the usage of this substrate for bioenergy production by anaerobic digestion (AD) is reviewed. An extended portrayal and a distinct appraisal are focused on different pretreatment methods to highlight their full potential for the production of bioenergy. Further innovative approaches for bioenergy generation are discussed.

The cost and energy assessment for the pretreatment of FW is elaborately discussed. Finally, the review concludes with a summary of the challenges and recommendations for further research and development.

## 2. Food Waste and Its Characterization

The upsurge in the population has led to an increase in demand for food and its related products since it is essential for life's existence. The consumption of foods occurs in a different manner for various levels of organisms. For instance, microbes consume food in the form of carbohydrates, vitamins, etc., whereas humans ingest fruit, pulses, cereals, etc. [4]. The improper management of these foodstuffs during their life cycle leads to the production of piles of FW that eventually causes social and economical issues [5]. About 1.3 billion tons of food are wasted per year globally which causes civic and political distress [6,7]. Moreover, according to the report by Zhongming et al. [8] about 8–10% of greenhouse gas emissions are caused due to leftover foods. It was estimated that about 26% of FW is generated in the drink industry, 21% in dairy processing, 14.8% in vegetable processing, 12.9% in cereal processing, 8% in meat processing, 3.9% in oil processing, 0.4% in fish processing, and 12.7% due to the other sources as per the study by Baiano et al. [9]. The composition of different food wastes is shown in Table 1.

The composition of FW is dependent on geographical regions and human food habits. Regardless of the inexorable production of FW, it is also rich in other nutritional components. It mainly consists of protein, lipid, cellulose, lignin, hemicellulose, and starch that collectively encompass 82–96% of volatile content [10]. Due to the presence of a higher amount of carbonaceous, protein, and lipid content, it is largely used for the production of bioenergy. Nitrogen, potassium, and phosphorus are also present in FW, which are necessary for plants for its reproduction; thus, it has greater potential while recycling it as a fertilizer [11,12]. The hydrolysis of protein and carbohydrates are higher compared to lipids as reported in the study by Xu et al. [13]. The lipid content was found to be lower in fruit and vegetable waste whereas it was higher in FW due to the existence of fats and oils [14]. The proximate and ultimate analysis of the different types of food waste in different studies is shown in Table 2. The proximate analysis provides the detail about the moisture content, total solids, and volatile solids in organic substances, which shows the waste nature of combustion [15]. The ultimate analysis provides an in-depth analysis of carbon, hydrogen, oxygen, nitrogen, and sulfur concentration in waste matter. In general, FW is rich in water content to about 80% by mass, thus making it liable for biological degradation when exposed for a longer period. Thus, it is necessary to reduce the water content by adopting a suitable pretreatment method to diminish the energy input during processing [16]. The C/N ratio of FW was found to be in the range of 12.7–28.87. Likewise, the carbon and ash composition in FW varied in the range of 5.78–22.8% and 2.5–14.3%, respectively, thus resulting in above 50% of the coefficient of variation. Generally, a higher carbon and volatile composition is needed for biofuel production through traditional thermochemical conversion methods [17]. The higher heating value (HHV) of FW such as meat and dairy production (HHV > 25.2 MJ/kg) is efficient to enhance the energy output during the digestion process. In FW the carbon content has been reported to be between 40.0 and 60.0%, hydrogen content 5.0–13.0%, nitrogen content 1.5–6.0%, and oxygen content between 17.0 and 41.0%. The sulfur content was found to be lower with a value of <1%. A higher carbon and hydrogen content increases the biomass calorific value and thus improves energy generation [18]. The curious fact about the FW is it is rich in nutrient and water content, thus generating organic pollutants that lead to the negative effect on environment [4]. FW is a feedstock that possesses zero or fewer gaining costs which contributes to the origination of the inventive model. By considering sustainable development, research has been undertaken to upgrade this waste to energy and valued products [19].

**Table 1.** Composition of food waste.

| Waste Type | Composition | Reference |
|---|---|---|
| Food waste | Fat: 8.79% of VS<br>Protein: 17.17% of VS<br>Carbohydrate: 74.04% of VS | [20] |
| Cafeteria food waste | Total carbohydrate: 277.2 $\pm$ 0.1 g/L<br>Total protein: 114.3 $\pm$ 0.4 g/L | [21] |
| Raw fresh food waste | Carbohydrate: 37.6%<br>Crude protein: 19.2%<br>Crude fat: 32.5%<br>Cellulose: 16.8%<br>Hemicellulose: 8.3%<br>Lignin: 8.7% | [22] |
| Food waste | Cellulose: 2.0 wt%<br>Hemicellulose: 1.2 wt%<br>Lignin: 0.1 wt%<br>Extractives: 96.7 wt% | [23] |
| Dried food waste | Protein: 7 wt%<br>Lipid: 10 wt%<br>Carbohydrate: 67 wt%<br>Insoluble dietary fiber: 5 wt%<br>Salt: >0.65 wt% | [24] |
| Food waste | Carbohydrate: 48%<br>Protein: 15.1%<br>Lipid: 10.6% | [25] |
| Food waste | Carbohydrate: 31%<br>Protein: 14.1%<br>Cellulose: 13.9 g/L | [26] |
| Food waste | Carbohydrate: 43.5% VS<br>Protein: 18.4% VS | [27] |
| Restaurant food waste | Crude fat: 31.8% TS<br>Cellulose: 4.70% TS<br>Hemicellulose: 10.05% TS<br>Lignin: 2.12% TS<br>Crude protein: 15.5% TS<br>Carbohydrate: 41.6% TS | [28] |

**Table 2.** Proximate and ultimate analysis of food waste.

| Food Waste Type | Proximate Analysis | | | | Ultimate Analysis | | | | | C/N Ratio | Reference |
|---|---|---|---|---|---|---|---|---|---|---|---|
| | pH | Moisture (%) | TS (%) | VS (%) | C (%) | H (%) | N (%) | O (%) | S (%) | | |
| Kitchen waste—highly acidic | 2–5 | 86 | 14 | 88 | 41.8 | 5.06 | 2.01 | | | 20.8 | [29] |
| Kitchen waste—medium acidic | 5–7 | 89 | 11 | 89 | 40.3 | 5.14 | 1.92 | | | 21 | |
| Kitchen waste—lesser alkaline | 7–8 | 90 | 10 | 95 | 42.1 | 5.21 | 1.86 | | | 22.6 | |
| Dried food waste | | 9.91 | | 70.2 | 43.1 | 6.91 | 3.14 | 38.45 | 0.62 | | [30] |
| Food waste | 5.4 | 0.97 | 20 | 18.01 | 50.69 | 7.35 | 3.51 | 0.28 | 0.42 | | [31] |
| Okra waste | 7 | | 15.15 | 13.11 | 39.3 | 5.39 | 3.21 | 35.74 | | 12.2 | [32] |
| Food waste | | 93.74 | 6.42 | 5.85 | 48.04 | 5.71 | 2.95 | | 0.44 | | [33] |
| Food waste | | 3.3% | | 71.3 | 47.5 | 6.6 | 3.9 | 31.3 | 0.4 | | [23] |
| Dried food waste | | 6.09% | | 80.9 | 41.5 | 5.76 | 1.55 | 51.04 | 0.12 | | [24] |
| Food waste | 5.1 | 14.8 ± 0.6 | 85.2 ± 0.6 | 44.3 ± 2.8 | 47.0 ± 2.4 | 7.3 ± 0.4 | 2.7 ± 0.4 | 42.9 ± 3.3 | 0.2 | 18:1 | [34] |

## 3. Impeding Components in Food Waste

FW contains abundant biomolecules such as proteins, lipids, and sugar. The presence of these components enhances its organic content for bioenergy production whereas it also acts as an inhibitor during the energy production process as shown in Figure 1. The presence of a higher lipid content increases the digestion time and causes system failure due to the formation of long-chain fatty acid (LCFA) during energy production [35]. This quicker acidification occurs due to the presence of carbohydrate content that affects the C/N ratio causing nutrient constraints. In some types of FW, the structure is more complex with higher lignin content and takes several days for hydrolysis whereas protein and lipids are hydrolyzed in fewer days. The lower C/N ratio and higher nitrogen content in FW lead to the generation of a higher amount of ammonia that causes inhibitory effects and process worsening during the energy generation process [36]. Ammonia is generated due to the degradation of protein and acts as a micronutrient for microorganism growth. The presence of ammonia during the digestion process causes toxicity and thus hampers the action of anaerobic microbes. The presence of free ammonia diffuses inside the cell and disrupts cellular activity. The presence of salt such as magnesium, calcium, potassium, and sodium and heavy metals such as chromium, cobalt, nickel, and zinc in higher concentrations disturbs the digestion process [37]. The salts are not easily degraded and this causes accumulation in the digestion tank whereas the heavy metal disturbs the activity of enzymes. The presence of these components greatly hinders the complete digestion operation.

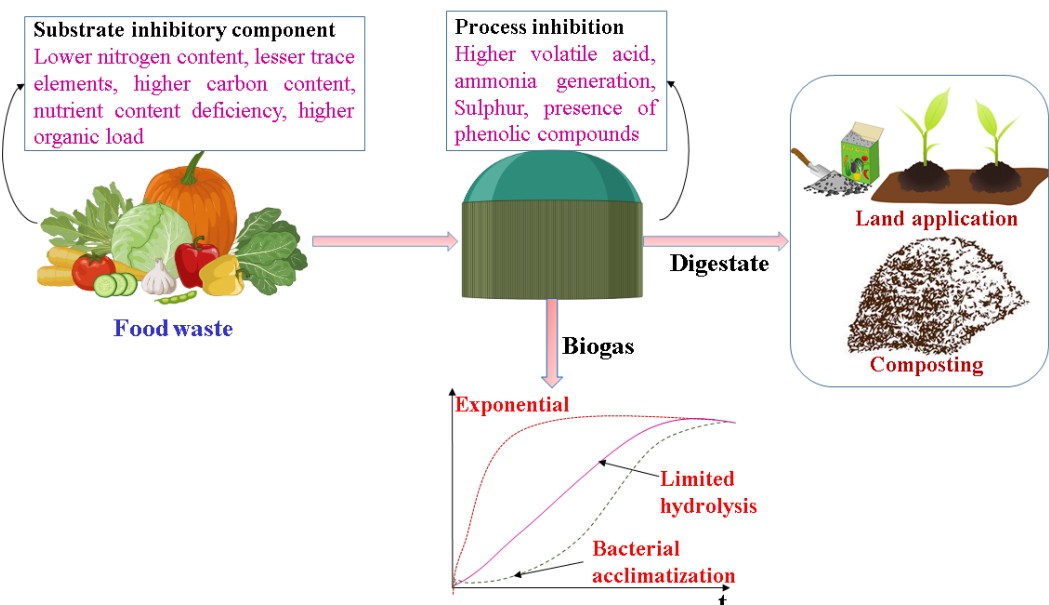

**Figure 1.** Inhibitory compounds affecting the digestion process.

## 4. Anaerobic Digestion of Food Waste

AD is a cost-efficient technology for the treatment of FW and the production of bioenergy since FW favors anaerobic microbial growth [13,38]. The presence of higher moisture content and biodegradability in FW serves as a better substrate for AD [39]. In AD, the FW organics are broken down by the means of microorganisms in anaerobic conditions. The biogas retrieved in this process is purified and in turn converted to electricity, heat, and power. In FW, the protein and carbohydrate are hydrolyzed at a faster rate whereas the lipids have a lower biodegradability but then also it enhances the quality of energy produced [40]. During AD, the degradation involves four major steps, namely, hydrolysis, acidogenesis, acetogenesis, and methanogenesis which are carried out by the group of bacteria as shown in Figure 2 [41].

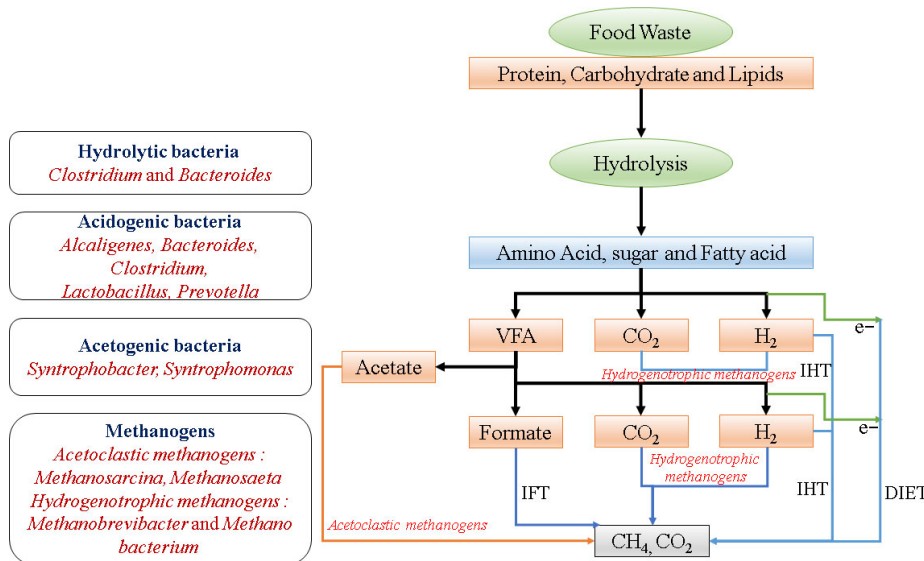

**Figure 2.** Mechanism of interspecies electron transfer in the anaerobic digestion process.

　　Different microbials are liable at every stage and have a syntrophic association with the microbes in other stages of digestion [42]. In the hydrolysis stage, the biomolecular components in FW are hydrolyzed to simpler monomers in the form of amino acids, sugar, and fatty acids. This stage is initiated by the hydrolytic enzymes that secrete exoenzymes such as amylase, cellulase, protease, lipase, etc., where these enzymes are adsorbed onto the surface of the substrate to reduce the components to simpler ones. Hydrolysis is the rate-limiting stage since the organic waste matter is highly recalcitrant. The second stage is the acidogenic phase, where the monomers are degraded to volatile fatty acid (VFA) by means of acidogenic microbes such as *Bacillus*, *Salmonella*, *Streptococcus*, *Escherichia coli*, and *Lactobacillus* [43]. Then, the VFA is converted to acetic acid and hydrogen by means of acetogens such as *Syntrophomonas* and *Syntrophobacter* [7]. The acetic acid produced is utilized by methanogens. A syntrophic relation was found between acetogens and hydrogenotrophic methanogens which, respectively, utilize acetate and hydrogen for the production of biogas [44]. The interspecies electron transfer (IET) between the syntrophic microorganisms assesses the efficiency of methane generation. It occurs in three different modes such as interspecies hydrogen transfer, interspecies formic acid transfer, and direct interspecies electron transfer. A stable IET between acetogens and hydrogenotrophic methanogen enhances methane production due to the transfer of electrons through a redox reaction between syntrophic microbes [45]. Organic acid with hydrogenase content uses a proton as an electron acceptor to oxidize organic components to hydrogen, which is further consumed by hydrogenotrophic methanogens. Interspecies formic acid occurs due to the oxidation of formate to $CO_2$ and then to methane due to formate dehydrogenase [46]. Direct interspecies electron transfer degrades VFA by utilizing cytochrome C, an iron heme protein, and other proteins as an electron transfer medium [47,48].

　　The last stage in AD is the methanogenesis process, where the products from the previous phase are converted to biogas employing methanogenic archaea [49]. Acetoclastic methanogens convert the acetic acid to methane and $CO_2$ whereas hydrogenotrophic methanogens consume hydrogen to generate methane [50,51]. Hydrogenotrophic methanogens such as *Methanospirillum hungate* and, *Methanoculles receptaculi* maintain a lower hydrogen partial pressure of less than 10 Pa, which is essential for the metabolic action of acetoclastic methanogens. It grows much faster than acetoclastic methanogens such as *Methanosarcina thermophile*. The methane production is affected by the pH value since the growth of acidogens and methanogens grows at 7–7.5 pH [52]. The AD process suffers due to the inhibitor formation, temperature, pH monitoring, and organic load concentration as studied by Opatokun et al. [18]. The recalcitrant compounds present in the substrate

negatively affect the bacterial growth and activity which results in VFA accumulation and thus decreases the pH, which further restricts the degradation process. Thus, in this entire process, hydrolysis is considered to be the major rate restricting due to the presence of complex biopolymers and thus a pretreatment is necessary to cope with this issue [53].

## 5. Existing Pretreatment Methods

In FW, the hydrolysis and methanogenesis are considered to be rate-determining steps during AD due to the presence of both easily degradable matter such as carbohydrate and non-degradable matter such as lipids, protein, and lignin components [54]. The efficiency of hydrolysis is highly subject to the substrate nature, its concentration, and the temperature of the digester [55]. During the biochemical reaction, the optimum condition for biogas production is necessary since the condition optimization leads to mass transfer of reaction, and apart from that the perfect feedstock is also necessary [56]. Pretreatment is the supplementary stage that enhances the availability of substrate to the anaerobic microbes during the digestion process [42]. The results are greatly influenced by the nature of the substrate and pretreatment mechanism [57]. The aim of the pretreatment is substrate size reduction, reducing the complex polymers to simpler compounds in order to enhance the fermentation step, and reducing the substrate crystallinity [58]. Karthikeyan et al. [59] stated that the pretreatment enhances biogas production with the improvement in biomolecule hydrolysis rate kinetics. Moreover, the reduction in toxic components occurs, which interrupts the AD process and thus minimizes the sludge quantity. The different categories of pretreatment are physical (thermal and microwave), chemical (acid or alkali), mechanical (sonication), and biological (enzyme and bacteria) as depicted in Figure 3 [60–65].

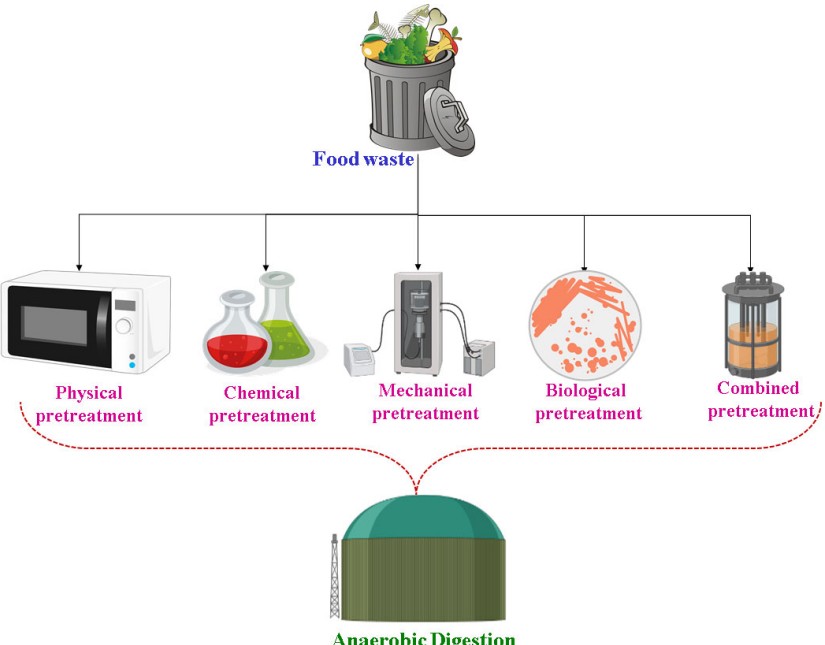

**Figure 3.** Pretreatment methods to enhance the AD process.

### 5.1. Physical Pretreatment

Physical pretreatment emphasizes particle size reduction and modifies the morphology of the substrate which thus enhances soluble organic release (SOR) [66]. It includes thermal and microwave pretreatment.

The thermal method enhances the surface area of the substrate thus improving the contact between the substrate and microbes, thus resulting in enhanced biogas production. The substrate is heated at a lower or higher temperature to deaggregate the cytomembrane [49,67]. At a higher temperature condition, the microbes in FW are reduced thereby

leading to lower biogas production. Moreover, a higher temperature >150 °C leads to the generation of inhibitory compounds such as melanoidins, which affects the AD process as well as not being economically and eco-friendly [59]. Liu et al. [68] studied the effect of kitchen waste, fruit or vegetable waste, waste-activated sludge, and municipal waste. Out of these, the solubilization increased to 59.7% and 58.5% in kitchen waste and fruit or vegetable waste during thermal pretreatment at 175 °C. In contrast to the solubilization, the methane yield was decreased by 7.9% and 11.7% in kitchen waste and fruit or vegetable waste due to the meladonin production at a higher temperature. Ariunbaatar et al. [69] reported that about 52% higher methane was produced at low-temperature thermal pretreatment of 80 °C at 1.5 h. Likewise, Gnaoui et al. [70] studied the effect of temperature of 100 °C on FW and found that the solubilization enhanced to 43.4% which resulted in a 23.68% improvement in methane yield than that of untreated FW. It was confirmed in various studies that the low-temperature treatment enhances the disintegration of FW with no inhibitory complex production [71].

The microwave (MW) method was adopted with the electromagnetic energy of 300 MHz–300 GHz, which by the dipole force enhances the kinetic energy and reaches boiling point [72]. In FW, the MW absorption leads to the vibration and generation of more heat, thus causing the solubilization of biopolymers [73]. Microwave-mediated pretreatment on FW shows a higher accumulation of propionic acid and thus causes inhibitory action [74]. Sun et al. [75] adopted MW–calcium hydroxide pretreatment on kitchen waste and found enhanced protease action for the methane generation of 430.4 N mL/g VS. It was reported in the study by Shahriari et al. [76] that the MW pretreatment is not efficient for hydrolysis for easily degradable waste.

### 5.2. Chemical Pretreatment

The chemical pretreatment method is a widely used method where the solubilization of complex organics takes place for easy accessibility by microbes and reduced digester HRT [77]. Different types of chemical pretreatment, namely, acid, alkaline, ozonation, and advanced oxidation process (AOP) are employed methods for biomass disintegration [78,79].

Acid pretreatment is carried out with concentrated or dilute acids such as hydrochloric acid, sulfuric acid, and citric acid [80]. Vavouraki et al. [81] used hydrochloric acid of 1.12% and 1.17% at 100 °C to enhance the soluble sugar concentration of 120% as compared to untreated FW. Generally, concentrated acid pretreatment is avoided due to the generation of inhibitory components such as hydroxymethyl furfural, carboxylic, phenolic, and furfural compounds [82]. Dilute acid is highly preferred due to its reduced toxicity and fewer corrosive and lesser reagents are used [83]. In contrast, Karthikeyan et al. [84] reported that the carbohydrate is not easily hydrolyzed by acid pretreatment alone and it leads to a decrease in biogas yield by 45%.

Alkali pretreatment is efficient in solubilizing the lignin content where it eradicates the feedstock acetate structure, making it accessible for hydrolytic enzymes [79]. The widely used alkalis are sodium hydroxide, potassium hydroxide, and calcium hydroxide and they help in improving the performance of AD [85,86]. In a study by Linyi et al. [87] the calcium oxide pretreatment of FW showed the maximum biogas yield of 829 mL/g VS, which shows the enhancement in solubilization. In some reagents, the released cations such as calcium and magnesium show an inhibitory effect and it cannot be neglected since it harms methanogenic archaea as per the study by Ariunbaatar et al. [88]. They reported the calcium and magnesium concentration of 0.2 and 0.72 g/L for microorganism activity. It was confirmed in many studies that the acidic condition leads to a lower biogas yield than the alkaline condition.

Ozonation is another type of chemical pretreatment method where the ozone is employed for the solubilization of the substrate. The major advantage of the pretreatment method is no odor generation due to pathogens present in the substrate. It occurs through two actions, namely, oxidation and hydroxyl radical production. No complete oxidation

of the substrate takes place by the action of ozone. The substrate is broken down into dispersed particles and then the particles are diffused with soluble organic to aqueous medium [89]. It is efficient for the substrate with a very complex structure whereas it is not very efficient for FW due to the loss of fermentable sugar and thus reduces biomethane production [1]. Sethupathy et al. [90] adopted an ozonation pretreatment on fruit waste and the emancipation of soluble organics of 17.6% was reported at the ozone dosage of 0.04 g/g SS. Further combining the ozone with citric acid of 0.03 g/g SS yields a soluble release of 24.4%, which is efficient for energy generation.

AOP is a chemical method where the oxidation of biomass takes place. The most common type of AOP is the Fenton process, where the $H_2O_2$ reacts with ferrous ions at an acidic pH to generate highly oxidative radicals. This method does not produce any inhibitory components and to date has been widely applied for sludge pretreatment [91]. Its effect on FW has barely been studied. Magare et al. [92] reported that the de-oiled biomass of citrus fruit waste after the extraction of essential oil had the ability to produce biogas and biomethane of 322.6 and 122.4 mL/g VS at a 30% Fenton dosage.

### 5.3. Mechanical Pretreatment

The mechanical pretreatment method is aimed at reducing the particle surface and enhancing the surface area of organic matter and solubilizing it to release to the aqueous phase [49]. The enhanced surface area leads to efficient contact between the substrate and anaerobic microbes, which in turn improves biogas production in the AD process. Considering the economic point of view, the mechanical pretreatment operation cost is reduced when combined with physical or chemical methods [93]. The mechanical pretreatment widely used for FW disintegration is the ultrasonication method.

Ultrasonication pretreatment works by generating shear force, cavitation, pressure drop, and radical formation caused due to higher-intensity ultrasonic waves that disrupt FW and thus improve the reaction rate by reducing retention time [94]. Rasapoor et al. [95] concluded that the biogas production by sonication pretreatment increased to 80% when the OLR was maintained at 1500 g VS/m$^3$. Li et al. [96] reported that the higher glucose yield was enhanced with a shorter sonic pretreatment time and thus the reactor size or enzyme requirement was reduced to greater than 50%. In a study by Yue et al. [97], the energy was found to be higher in ultrasound treatment by 18% than in the MW pretreatment. Moreover, it reduces acid accumulation and thus improves substrate consumption by anaerobic microbes. This pretreatment has been extensively implemented in the full-scale application of AD where the Sonix and Sonolyzer, Biosonator, MaXonics, and Hielscher are commonly applied systems [83].

### 5.4. Biological Pretreatment

A biological pretreatment is an eco-friendly approach which does not generate any inhibitory compounds. It is a slow process that entails a longer retention time and the microorganisms use the released sugar as a carbon source during pretreatment [84]. It has become a popular pretreatment method which includes microbes and enzymes that promote substrate hydrolysis and enhance the digestion rate. The microbes secrete an enzyme that is essential to degrade complex polymers. Commercially available enzymes are protease and amylase and they have been widely adopted in enzymatic pretreatment [98]. The protease and amylase enzyme degrades the protein and carbohydrate into amino acids and sugar [99]. The major characteristic of enzyme pretreatment is its efficiency which shows a better dissolution of organic matter and thereby enhances methane generation [100]. Han et al. [101] reported that the waste hamburger enzymatic hydrolysis was enhanced by commercial amylase where the enzyme loading of 0.14 mL/L enhanced the hydrolysis efficiency of 0.784 g reducing sugar/g substrate. Likewise, Liu et al. [102] reported the enhancement in the hydrolysis of waste pizza was achieved at the amylase dosage of 0.02 mL/L. The waste hydrolysis with the pure microbial culture- secreting enzymes have the ability to yield more sugar whereas the cost increases to about 25% of the total cost [103].

The hydrolysis of protein by protease leads to the generation of ammonia, which leads to FW acidification and pH neutralization [104]. The lipid present in FW is hydrolyzed by lipase enzymes. The fat oil grease from FW was investigated by Meng et al. [105] for methane production by lipase I and lipase II enzyme pretreatment at 24 h with the dosage of 1000–5000 µL and an increase in methane yield in fat oil and grease by 157.7%, 53.8%, and 40.7% was found which moreover decreased the digestion time.

*5.5. Combined Pretreatment*

Combined treatment is adopted to reduce the negative effect and cost barriers in a single pretreatment method and to enhance the efficiency of pretreatment for AD. The major advantage of this pretreatment method is the enhancement of solubilization by applying two or more pretreatment methods with very minimal energy usage. Generally, the mechanical pretreatment along with chemical and physical pretreatment enhances the FW properties prior to AD. The surfactant shows higher lipid solubilization, substrate conversion, and reduced surface tension in the medium [106]. Shanthi et al. [107] adopted a combined pretreatment of fruit and vegetable waste using dimethyl sulfoxide and sonication at 90-watt sonic power and surfactant dosage of 0.008 g/g SS led to an enhancement in the delignification by 70% using surfactant initially and a higher solubilization of cellulose of 22% was achieved after combined pretreatment. This in turn led to the enhancement in methane production of 190 mL/gCOD. Likewise, the surfactant sodium dodecyl sulphate (SDS)-aided sonication pretreatment enhanced the organic release of 26% at 0.035 g/g SS SDS dosage and the sonic specific energy input of 5400 kJ/kg TS which led to a methane yield of 0.6 g/g COD [108]. Among different biopolymers, the lipid was described to be highly impacted during AD due to the formation of a waterproof layer that required higher energy and restricted the activity of methanogens [109]. Ravi et al. [110] used four different surfactants, namely, SDS, rhamnolipid, glucopon, and triton X-mediated sonication treatment on mixed FW to assess its hydrolysis potential. The triton X of 0.01 g/g SS along with sonication specific energy of 580 kJ/kg TS showed a higher solubilization of 45.5%, which in turn increased the carbon efficiency of 83% as compared to other surfactants.

Karthikeyan et al. [59] concluded that the amalgamation of mechanical and physical or chemical pretreatments was a better option for FW pretreatment whereas Ma et al. [66] and Zou et al. [53] concluded that the enzyme-mediated pretreatment was the best option for soluble organic discharge with a higher solubilization efficacy. The FW pretreatment negatively linked with methane generation due to the optimization of different parameters such as the selection of pretreatment methods and its optimization and the nature of substrate. Moreover, there is no standardization of the pretreatment methods for mixed FW as described by Karthikeyan et al. [59]. After the pretreatment process, the post treatment of FW also needs to be developed in upcoming years in order to accomplish the process in efficient manner and moreover, the process integration is also necessary as suggested by Kannah et al. [15].

## 6. Advantages and Limitations of the Pretreatment Method

Presently, different pretreatment methods have been developed to optimize the efficiency of hydrolysis and to select the suitable methods and operating conditions based on the nature of biomass [111]. The advantages of different single pretreatment methods are combined to enhance the hydrolysis efficiency and this reduces the negative impact of a single pretreatment. While considering the single pretreatment method, the physical method ensures greater stability and increases surface area by degrading the structure. In the chemical method, the partial solubilization of lignin and the higher solubilization of cellulose takes place and its capital cost is low. Mechanical methods are efficient at reducing bulk FW and the separation of unwanted objects. Biological pretreatment has lower operational costs since no chemicals are required. Moreover, AD performance is enhanced since no inhibitory compounds are produced [7]. Accurate implementation of the

pretreatment method for FW is vital since the efficacy of selected pretreatment describes the process profit and thus yields bioenergy production.

Despite all the advantages of the pretreatment method, there are many disadvantages in these approaches. In the chemical method, the process requires a higher operating cost and forms many inhibitory compounds such as carboxylic acids, furans, and phenolic compounds, and requires a longer retention time. Moreover, it generates more irretrievable salts [112]. The physical pretreatment has a greater limitation in its biodegradation due to the production of intermediate compounds by the Maillard reaction and thus has a higher capital cost due to higher energy and chemical requirements [113]. Mechanical pretreatment requires higher start-up costs due to the higher requirement of electricity and pathogen removal is also not very significant. Biological pretreatment has a higher enzyme cost and needs more process time.

### 7. Energy and Cost Assessment of the Pretreatment Sector

The energy aspect is the necessary factor to be assessed for implementing the pretreatment method conducted in the lab to an industrial scale. The optimization of the pretreatment method is essential between biogas generation and energy usage during the pretreatment method. The economic feasibility of the pretreatment method is directly impacted due to the energy balance. The sustainable pretreatment method is achieved by improving the output energy in the form of biogas as that of the input energy. The major factors that restrict the application of pretreatment methods are their higher cost, more energy usage, costly chemical usage, and operating conditions optimization.

Pretreatment methods such as ultrasonication, microwave, and disperser use higher electrical energy, and the energy spent during this method is calculated as

$$EE = P \times T/V \times TS \tag{1}$$

where EE is the electrical energy in kJ/kg TS, P is the power used during sonication or dispersion in kW, T is the time in sec, V is the volume of sample in L and TS is the total solid concentration in $kg/m^3$.

In the physical pretreatment method, heat is required as an energy source and the thermal energy incurred for pretreatment is calculated as

$$TE = M \times SH \times (T_2 - T_1) \tag{2}$$

where TE is the thermal energy in kJ, M is the mass of FW in kg, SH is the specific heat of FW in kJ/kg °C, and $T_2$, $T_1$ are the final and initial temperature in °C.

In the chemical and biological pretreatment method, the energy consumption is very low and makes the pretreatment feasible. It considers stirring energy alone and it can be calculated as

$$SE = P \rho \eta D \tag{3}$$

where SE is the stirring energy in kW, P is the power number of impeller, $\rho$ is the density of FW in $kg/m^3$, $\eta$ is the revolution per second and D is the diameter of impeller in m.

A study by Kavitha et al. [67] adopted a chemo thermo disperser on FW for biomethane production and found that the increase in liquefaction from 50 to 60% increased the energy required for the pretreatment process and an insignificant improvement in output energy was observed. The positive net energy of 224.16, 245.7, and 266.3 kWh was observed at 40–60% liquefaction of FW with the energy ratio of 1.12, 1.13, and 1.14. Although the energy ratio was greater at the solubilization of 50–60%, this did not contribute to methane production and profitability. Likewise, a study by Shanthi et al. [107] used a combined surfactant and sonication pretreatment on fruit and vegetable waste and found an energy ratio of 0.8, which is lesser than 1 whereas the biomethane generation was efficient due to a lower concentration of lignin. Liu et al. [114] used MW pretreatment on FW and sludge as a substrate for the AD process. It was found that the methane yield was higher with the

output energy of 76.25 kJ/g VS in MW pretreatment than that of the untreated biomass and the energy balance of MW showed 74.25 kJ/kg VS. The microwave pretreatment of yard waste followed by the co-digestion of FW and yard waste for bioenergy production was conducted by Panigrahi et al. [115]. The energy balance analysis showed a positive net energy of 6.5 kJ/g VS at the F/M ratio of 1. During the co-digestion of FW and sludge, the electric energy of 26.44 billion kWh was recovered each year [116].

Cost assessment is also an essential factor used to assess the economic viability of the pretreatment method, whereas it is still in the lab scale. The substrate hydrolysis was enhanced by pretreatment and the cost required for this pretreatment was assessed at a lab scale. The cost estimation was performed based on the difference between the cost used up and the annual return. The used-up costs were the investment cost, pretreatment cost, and cost of maintenance. The net gain is calculated based on energy sales and the cost used for pretreatment. Ariunbaatar et al. [69] used thermal energy to treat the FW for the production of biomethane and found a profit of 7.65–13.45 EUR/ton FW at the optimized pretreatment condition. Likewise, the chemo thermo disperser pretreatment on FW showed a net profit of 93 USD/ton FW with liquefaction of 40% as per the study by Kavitha et al. [67].

## 8. Different Innovative Approaches in the Field of Bioenergy Generation

Apart from different pretreatment methods that have emerged in different years, many limitations still persist due to their higher cost, inhibitory by-product formation, and higher energy requirements. To address these issues, certain innovative strategies have emerged for the enhancement of the availability of cheap substrates for biogas generation, as depicted in Figure 4. To address these issues, co-digestion, integrated energy production, and bioaugmentation strategies have been explored [117].

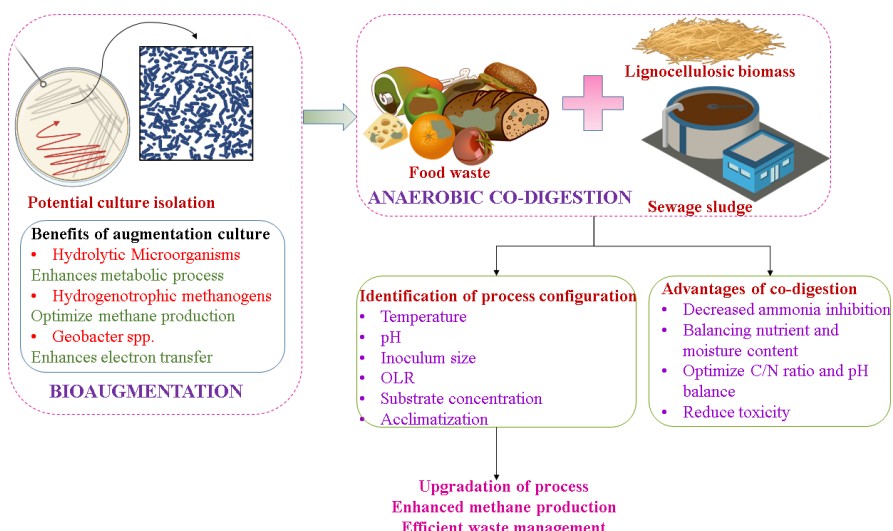

**Figure 4.** Innovative strategies for enhanced biogas production.

In a single digestion process, the lower biogas yield occurs due to recalcitrant and protein substrate and toxic components in feedstocks. These disadvantages can be enhanced by the anaerobic co-digestion (ACOD) process optimum mixing ratio. ACOD is the concurrent treatment of different substrates simultaneously in the same digester and inoculum. It helps to decrease the inhibition caused due to ammonia and enhances biogas production [118]. Different approaches are applied to increase ACOD efficiency, which significantly improves the economy by reducing operation and capital costs. The enzymatic pretreatment using amylase and cellulase was carried out for the ACOD of cow manure and corn straw as per the study by Wang et al. [119]. It was found that the methane yield was above 100%. The balance in the C/N ratio reduced ammonia inhibition. The balance of the C/N ratio to 32 showed a reduction in ammonia by 30% while co-digesting FW, paper,

garden, and fruit or vegetable waste [120]. Higher methane production was found while using co-substrate fruit or vegetable waste and cow manure and thermal pretreatment. This higher methane production was attributed to the higher C/N ratio due to the manure and thus decreased the toxicity due to ammonia [121]. Microwave-aided pretreatment efficiency on FW is affected due to acid accumulation and causes inhibition. It can be improved by co-digesting FW with sludge [74]. The major advantages of ACOD are an enhancement in process stabilization, restricted substance dilution, balancing the C/N ratio, methane enrichment, and an enhanced moisture content [122,123]. The rate-limiting step which is hydrolysis in AD is enhanced by co-digesting FW with sewage sludge since the sludge is rich in protein and causes inhibition. The simpler biodegradable matter FW enhances hydrolysis due to faster anaerobic bacterial growth. This improvement in microbial growth not only enhances hydrolysis but rather it improves the acidification and methanogenesis step [40]. The co-digested FW and sludge with the ratio of 0.5:0.5 showed higher methane productivity of about 4.59 times with the hydrolysis rate increment by about 3.88 times as per the study by Pan et al. [124]. Concerning the microbial population, the wide variety of microbes were used in ACOD system by familiarizing it continuously to the combined substrates. Widely used microbial populations are *Firmicutes*, *Bacteroidetes*, and *Proteobacteria* [125]. The accretion of inhibitory constituents leads to the augmented diversity in archaea and thus alters the acetoclastic methanogenesis to hydrogenotrophic methanogenesis [125,126]. ACOD has certain issues due to the occurrence of operational problems linked to raw material handiness and the complex nature of raw material due to the variation in degradability behavior caused by the parameters such as its nature and composition.

Integrated energy is a process where bioenergy retrieval is combined with the other valued products to overcome economic barricades [127]. The integration of biomethane and biohydrogen production was proposed to be an efficient strategy to enhance the sustainability and economic feature of biogas production since biohydrogen is a fuel with higher energy content [128]. Digestate recirculation helps in stable and efficient methane yield due to the increase in microbial density [129]. A two-staged phase system for FW was analyzed for biohydrogen and biomethane production as per the study by Algapani et al. [130]. The recirculation rate of 0.3 shows the stable generation of both hydrogen and methane with the production of 3 and 2.9 L/L/d. Likewise, in another study by Zhang et al. [131] the AD for FW and gasification of discarded waste fraction showed the co-production of hydrogen and methane through the thermal equilibrium model and a higher hydrogen content of 28.9% and a methane yield of 680 mL/g VS at a thermophilic temperature with no generation of methane in vegetable waste were found.

Bioaugmentation improves the microbial population which is necessary for biogas production by adding enzymes to the AD system [132,133]. It decreases the lag phase during AD and improves reactor performance and relieves toxic inhibition. In the AD system, the major inhibitory components accumulated during the digestion of FW are propionate and acetate, and these can be degraded by bioaugmentation with appropriate microbes [134]. Jiang et al. studied the effect of bioaugmentation on FW for AD efficiency and found that the bioaugmentation seed of 0.25 g/L/d for every three days enhanced the biogas production by 12-fold [135]. Bioaugmentation by methanogens enhances the performance of AD with a feedstock of a higher C/N ratio. An acetoclastic methanogenic archaea *Methanotrix* was employed by Li et al. [136] for the degradation of acetate which thereby enhanced biomethane production. Regardless of several benefits associated with bioaugmentation, there are several negative impacts on the AD process. While using fungus for bioaugmentation, the methanogens have an inhibitory effect and thus suppress biomethane production [137]. The efficacious bio augmentation depends on the type of microbes as well as the density of the inoculum. The lower density of the inoculum affects the adaptability to the newer environment and causes biomass washout.

### 9. Challenges Encumbering the Digestion Process

FW management is completed by a well-established AD process since FW consist of different biopolymeric component that varies based on geographic origin. The selection of the perfect pretreatment method for all types of FW is not viable. Thus, the appropriate pretreatment method is necessary based on the composition of FW. Apart from wide research, there is still a knowledge gap that restricts the full-scale application. The deficiency in the pretreatment process and its optimization leads to the production of a lower yield of biogas due to the generation of intermediate toxic substances, which is considered to be the major challenge. Apart from this, the other challenges are the higher financial cost, VFA accumulation, and process instability [13]. Moreover, the lack of an appropriate design for the reactor is considered to be the major design. For the efficient production of biogas, the characterization of FW, the action of microorganism and archaea, enhancement in methane generation, and nutritional stability are considered to be the major challenges [138].

### 10. Future Prospects and Recommendation

- The sorting of FW from municipal solid waste is the major obstacle and it is not practiced in many countries. In the future, proper steps will be needed to separate FW from municipal solid waste to enhance energy production;
- The AD efficiency is enhanced and FW degradation is enhanced by different pretreatment methods. The optimization of the process is needed in most research but is also essential to characterize the FW structure. It is perceived that most of the pretreatment studies are narrowed to the lab scale and not to the pilot scale. From a long-term development point of view, future research must be more focused on a full-scale application to optimize the results based on energy and mass balance analysis;
- The stability of the AD of FW is an essential concern. The co-digestion method is now focused on by many researchers for biogas production from FW. Moreover, the feeding of FW as a co-substrate helps in eradicating the higher capital cost in existing industrial plants. Thus, the optimization of the co-digestion of FW is necessary. The adoption of the biorefinery concept by producing valued product along with biogas simultaneously is conceptual research and this topic needs future attention [49];
- Physical pretreatment methods are applied in large-scale applications whereas the energy requirements and maintenance charges are high and generate inhibitory components. These methods are present in large-scale applications whereas there are no clear data about economic analysis. In the chemical method, the biogas yield can be improved whereas it is widely applied in bioethanol generation and information is lacking about the AD process, and it is needed in further studies [11]. The combination of the pretreatment process shows many advantages and thus contributes to the favorable field of research to explore. Research must progress to understand the mechanism of combined pretreatment and a techno-economic analysis is needed to evaluate these treatment methods on an industrial scale.

### 11. Conclusions

Biogas production through the AD process is a favorable technique for FW management. However, some challenges such as inhibitory components in feedstock, complex feedstocks, low process stability, and higher acidification have led to very limited application of this process. To improve biogas generation, the optimization of operational parameters, pretreatment, or certain innovative methods have been studied. The non-pretreatment methods such as co-digestion and bio-augmentation of FW enhance the stability of the AD process, thus achieving higher energy production. Yet, further studies are necessary to develop more economic processes considering the biorefinery approach.

**Author Contributions:** P.M., G.M. and R.B.J.: conceptualization and methodology; P.M.: writing—original draft; G.M. and R.B.J.: writing—review and editing; G.M. and R.B.J.: supervision. All authors have read and agreed to the published version of the manuscript.

**Funding:** This research received no external funding.

**Data Availability Statement:** Not applicable.

**Conflicts of Interest:** The authors declare no conflict of interest.

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
