# Peer review of "Efficacious Utilization of Food Waste for Bioenergy Generation through the Anaerobic Digestion Method"

_processes, doi:10.3390/pr11030702_

Round 1

Reviewer 1 Report

This manuscript intends to present the state of the art in the characterization of food waste and its valorization through anaerobic digestion, and to present some pre-treatment methods to improve the latter. A summary of challenges and future recommendations was also made.

Following, I listed some remarks and specific comments that must be considered before publication.

General comment:

The work previously published by Atelge et al. (2020), which includes one of the co-authors of this manuscript, already aimed a more detailed review of pre-treatment technologies to improve anaerobic digestion. In that publication, food residues were also mentioned. The detailed characterization of food waste that the authors have included in this manuscript is quite important, but in my opinion the authors should present the main difference between the two reviews or justify the importance of a new publication on food waste pretreatment technologies to follow the previously published review.

Specific comments:

Line 58: Reference “UNEP, (2021)” should be replaced by [reference number].

Line 76:The proximate analysis provides the detail about the moisture, ash, and carbon contents in organic substances whereas the ultimate analysis provides the in-depth analysis of carbon, hydrogen, oxygen, nitrogen, and sulphur concentration in waste matter.”

What do you mean about ash and carbon contents? Do you refer to TS and VS analysis? This sentence is not correct. Also, oxygen and sulphur do not appear in Table 2.

Table 1: What does "S No." mean? Please specify.

Line 132: Please correct “microorganisms, bacteria and archaea,” instead of “bacteria”.

Figure 2: The most dominant bacteria populations found in the first stage (hydrolysis) belong to the phyla Firmicutes and Bacteroidetes, for example, genus Clostridium and Bacteroides, respectively.

Commonly, acidogenic bacteria found in AD have been assigned to members of genus Alcaligenes, Bacteroides, Clostridium, Lactobacillus, and Prevotella. As in the previous cases, the authors can indicate the dominant methanogens in AD. For example, Methanosarcina and Methanosaeta for acetoclastic methanogens, or Methanobrevibacter and Methanobacterium for hydrogenothrophic methanogens. To obtain this information, it is suggested to consult Table S1 of Supplementary Material in “Le Zhang, Kai-Chee Loh, Jun Wei Lim, Jingxin Zhang. 2019. Bioinformatics analysis of metagenomics data of biogas-producing microbial communities in anaerobic digesters: A review. Renewable and Sustainable Energy Reviews, 100, 110-126, https://doi.org/10.1016/j.rser.2018.10.021.”

Figure 2 and all the manuscript: Please correct “Methanogenic archaea” or “methanogens” instead of “Methanogenic bacteria”.

Line 132: Please rephrase “This stage is initiated by hydrolytic bacteria that secrete exoenzymes”

Line 141: Enzymes are adsorbed to the the surface of what?

Line 144: Please correct the microorganisms list according to the suggestion mentioned for Figure 2.

Line 147: Please correct: “…was found between acetogens and hydrogenotrophic methanogens which, respectively, utilize acetate and hydrogen, for the production of biogas.”

Line 161: Please correct “archaea” instead of “archae”.

Line 186: What does substrate behavior mean? Is it not the composition of the substrate? What's the difference?

Line 186: Please correct: “The aim of the pretreatment is:” instead of “The aim of the pretreatment are:”

Line 192: Please correct: “physical” instead of “Physical”

Line 217: Please delete the term “mesophilic” in this sentence. The mesophilic temperature applies to a range from 20 to 45°C, and this term applies mainly to the growth of microorganisms.

Lines 420 and 425: Please correct “methanogens” instead of “methanogenic microbes”.

Line 439: This is the first time that authors mention the disperser as a method. It is analysed in this section, but it was not considered previously as a pretreatment method in the manuscript.

Line 495: Please correct: “microorganisms” instead of “bacteria”. The production of biogas and methane involves the action of bacteria and archaea.

Line 528: Please rephrase: “pre-treatments or some innovative methods have been studied.”

Reviewer 2 Report

This manuscriptEfficacious utilization of food waste for bioenergy generation through anaerobic digestion methodfirst describes the source, composition, and characteristics of food waste(FW) and its advantages as a bioenergy source, and then describes the impeding components in FW that pose obstacles to processing. Then, anaerobic treatment, the most commonly used treatment method for FW was introduced, and the inhibition stage-hydrolysis, was pointed out. Then leads to the emphasis of this article-pretreatment technology, this part is detailed, including a variety of methods, the advantages and disadvantages of pretreatment and pretreatment cost evaluation. Subsequently described some innovative approaches, such as digestion fluid recycling and co-digestion, and finally discussed the challenges and future prospects about FW. It is a good review of FW processing with complete structure and detailed content. There are some recommendations are outlined below.

1.    1.1 Food waste and its characterizationshould become a single paragraph,not belong to1 Introduction

2.    Paragraph 6Different innovative approaches in the field of bioenergy generationand paragraph 7Energy and cost assessment of pretreatment sectorshould reverse the order,only in this way the article could be more logical.

3.    “The co-digestion” in Paragraph 6 could be a key point in this paragraph,becausethe co-digestionis the current research hot spot.

Round 2

Reviewer 2 Report

I have read both the original paper and the revise paper, and found the author have already done a lot of revise, which made the paper more readable. Now, this manuscript can be accepted as it is.

Author Response

Thank you for accepting.